# Lessons to Learn for Adequate Targeted Therapy Development in Metastatic Colorectal Cancer Patients

**DOI:** 10.3390/ijms22095019

**Published:** 2021-05-09

**Authors:** Helena Oliveres, David Pesántez, Joan Maurel

**Affiliations:** 1Translational Genomics and Targeted Therapeutics in Solid Tumors Group, Medical Oncology Department, Hospital Clinic of Barcelona, IDIBAPS, University of Barcelona, 08036 Barcelona, Spain; 2Gastrointestinal and Pancreatic Oncology Group, Hospital Clínic, IDIBAPS, CIBERehd, University of Barcelona, 08036 Barcelona, Spain

**Keywords:** insulin-like growth factor receptor, metastatic colorectal cancer

## Abstract

Insulin-like growth factor 1 receptor (IGF1R) is a receptor tyrosine kinase that regulates cell growth and proliferation. Upregulation of the IGF1R pathway constitutes a common paradigm shared with other receptor tyrosine kinases such as EGFR, HER2, and MET in different cancer types, including colon cancer. The main IGF1R signaling pathways are PI3K-AKT and MAPK-MEK. However, different processes, such as post-translational modification (SUMOylation), epithelial-to-mesenchymal transition (EMT), and microenvironment complexity, can also contribute to intrinsic and acquired resistance. Here, we discuss new strategies for adequate drug development in metastatic colorectal cancer patients.

## 1. Insulin and the IGF Pathway

Insulin and insulin-like growth factor 1 (IGF1) are closely related ligands that interact with the insulin receptor (INSR) and IGF1 receptor (IGF1R) family of receptors. Insulin is considered to be a metabolic hormone activating INSR, whereas IGF1 is generally considered to be a growth factor activating IGF1R. IGF1R is a homodimeric, type II receptor tyrosine kinase comprising two extracellular α subunits and two β subunits. The extracellular α subunits are required for IGF1, IGF2, and insulin ligand binding, whereas the two β subunits contain an extracellular portion, transmembrane region, and intracellular tyrosine kinase catalytic domain and ATP-binding site. IGF1R is closely related to INSR, with sequence similarity varying from 41% to 84%, depending on the domain. The interaction of the ligands with IGF1R, INSR, or hybrid receptors activates numerous downstream pathways within the cell. Ligand-activated IGF1R first binds to intracellular adaptor proteins, predominantly insulin receptor substrates (IRS1–6) and Src homology 2 domain-containing transforming protein 1 (SHC). Subsequent phosphorylation of these proteins induces the activation of phosphoinositide 3-kinase (PI3K) and mitogen-activated protein kinase (MAPK) pathways [1,2,3]. In addition, IGF1R signals different nuclear transcription factors that contribute, through signal transducers and activators of transcription (STAT) proteins [4], to epithelial-to-mesenchymal transition (EMT), cell survival via upregulation of anti-apoptotic Bcl-XL [5], and adaptive metabolic reprogramming via promotion of glycolysis and glutamine consumption through MYC activation [6].

The bioavailability of IGF ligands is abnormally high in many cancers, and IGF-binding proteins (IGFBPs) and IGFBP proteases are important for regulating ligand bioavailability [7]. Whereas IGF1R mutations have not been described in tumors, IGF1R mutations have been described as rare causes of intrauterine and postnatal growth disorders [8,9]. Polymorphisms in genes encoding either IGF1 or IGFBPs contribute to up- or downregulation of IGF1R function [10]. Upregulation of IGF1R and/or INSR constitutes a common paradigm mediating tumor resistance [11,12] in different cancer types and is associated with poor prognosis in prostate cancer, squamous cell lung cancer, and prostate cancer [13,14,15].

## 2. Mechanism of Resistance Related to the IGF1R Pathway

### 2.1. IGF1R Resistance Due to Other Tyrosine Kinase Receptors and Metalloproteinases

IGF1R interacts with tumor suppressor genes and proto-oncogenes. Disturbances in components in the p53/MDM2 network may upregulate IGF1R and confer cancer cells with a growth advantage [16,17,18]. Compensatory signaling mediated by IGF2 through the INRS can contribute to resistance to IGF1R compounds [19]. IGFBP proteins are tightly regulated by serine proteases and matrix metalloproteinases, particularly matrix metalloproteinase-7 (MMP7) [20,21,22,23], which is associated with poor prognosis in metastatic colorectal cancer (mCRC) [24,25]. MMP7 can indirectly contribute to IGFBP degradation [26] and IGF1R-related chemoresistance through AKT-dependent [27] and -independent pathways [28].

Other relevant aspects related to IGF1R signaling involve interactions with other oncogenic tyrosine kinase receptors and mutations in downstream pathways, such as PI3K/AKT and RAS/MAPK [29]. Crosstalk between IGF1R and other RTKs, including epidermal growth factor receptor (EGFR), epidermal growth factor 2 receptor (HER2), VEGFR2, PDGFR, MET, and ALK, results in reciprocal compensatory mechanisms that limit the response or mediate acquired resistance to IGF1R therapies [30]. One example of this IGF1R-EGFR crosstalk has been recently described in acquired osimertinib resistance in non-small-cell lung cancer [31].

### 2.2. IGF1R, SUMOylation, and Other Post-Translational Procesess

SUMO (small ubiquitin-like modifier) proteins are proteins of approximately 12 kDa in size with a structure similar to ubiquitin [32] that belong to a broader family of eukaryote-specific peptidic post-translational modifiers called ubiquitin-like (Ubl) proteins. SUMOs are present mainly in three isoforms in mammalian cells (SUMO-1 to -3; herein, collectively called SUMO), of which SUMO-1 is the most extensively studied, plus two other paralogs (SUMO-4 and -5) [33]. SUMOylation is mainly observed in nuclear and perinuclear proteins. SUMOylation can occur on both isolated proteins and those in supramolecular complexes where its molecular effects are often more difficult to decipher [34]. SUMOylation is a transitory and dynamic process in which SUMO proteins attach to lysine residues on targeted proteins. Only a small fraction of the targeted substrate is modified at any given time. Therefore, SUMO proteins are necessary to initiate a certain activity but not to maintain it [35].

There is functional crosstalk between the conjugation of SUMO to a substrate and other post-translational modifiers, mainly, ubiquitylation, acetylation, and phosphorylation. The interaction between ubiquitylation and SUMOylation is complex and SUMO and ubiquitin are often conjugated to the same substrate, lysine, acting antagonistically or sequentially [36]. Another post-translational modification that was recently identified as a SUMO regulator is acetylation. Acetylation of Ubc9 at residue K65 regulates the SUMOylation of some substrates [37]. There is also a complex interaction between SUMOylation and phosphorylation [38]. Phosphorylation of SUMO substrates can positively or negatively regulate their modification by SUMO, depending on the protein. On the other hand, SUMO can also regulate the phosphorylation system through the SUMOylation of kinases and phosphatases. SUMO proteins control cellular signaling networks and regulate, among others, DNA repair, differentiation, apoptosis, nuclear transport, and EMT differentiation [39,40,41,42,43].

Phosphorylation of IGF1R and its subsequent nuclear translocation has been reported to be mediated by SUMOylation [44]. IGF1R exhibits a different function when phosphorylated in the nucleus because it acts as a transcription regulator (instead of a tyrosine kinase receptor) and its nuclear location is associated with poor prognosis [45]. RANBP2 and PIAS3, the main SUMO E3 ligases, have been implicated in IGF1R nuclear accumulation [46,47]. In the nucleus, pIGF1R has AKT/MEK-independent properties, such as cyclin D1 and axin2 activation [48,49]. Interestingly, treatment of oxaliplatin-resistant colorectal cancer (CRC) cells, but not naïve cells, with IGF1R inhibitors (either ganitumab, a monoclonal antibody, or AEV-541, a tyrosine kinase inhibitor) promoted IGF1R nuclear internalization and PIAS3 inhibition decreased cytoplasm–nuclear IGF1R trafficking [47].

### 2.3. IGF1R and EMT

EMT is a well-known mechanism of intrinsic [50,51] and acquired [52] chemoresistance. IGF1R signaling was recently implicated in EMT induction/maintenance through STAT3/NANOG/Slug [53] but also, importantly, was found to contribute to acquired chemoresistance [54,55]. Although IGF1R and EMT markers have been shown to be upregulated in drug-tolerant cells, the exact mechanism of resistance remains unknown. In addition, IGF-alternative mechanisms of EMT induction include NF-KB activation [56], WNT/β-catenin [57], and TGFβ/SMAD activation [58], extensively reviewed by Li et al. [59]. Therefore, because other alternatives that are at least partially IGF1R-independent promote EMT, combinations of EMT inhibitors with IGF1R inhibitors are more likely to be effective in CRC.

### 2.4. IGF1R and the Microenvironment

The tumor microenvironment includes cancer-associated fibroblasts (CAFs) and vascular and immune cells. CAFs and myeloid cells contribute to STAT3 activation in cancer cells through IL-6, CCL2, TGFB, and CCL5 release, and promote regulatory T-cell (Treg) expansion [60]. Preclinical data concerning the combination of IGF1R and STAT3 inhibitors, but not IGF1R inhibition alone, showed a reduction in tumor burden through CAF and myeloid cell depletion [60,61]. Interestingly, anti-IGF1R therapies (cixutumumab) or radiotherapy can promote an immune-suppressive microenvironment due to IGF2- [61] or CAF-dependent IGF1 release from cancer cells [62]. In this latter work, radiotherapy-activated CAFs promoted the survival of CRC cells through metabolic reprogramming (favoring increased glycolysis and glutamine consumption). Therefore, this seminal paper indicates that paracrine IGF/IGF1R signaling contributes to metabolic reprogramming (induction of lactate and glutamine consumption) [63,64,65], a well-known mechanism in the immune-suppressive tumor microenvironment, due to M2 polarization [66] and Treg selection [67].

## 3. IGF1R Pathway: Lessons to Learn for Adequate Drug Development in *RAS* Wild-Type mCRC Patients

### 3.1. IGF1R in the Clinical Scenario

Several studies have evaluated the activity of the IGF1R receptor or its ligands (either IGF1 or IGF2) or IGFBPs as a surrogate marker of chemotherapy or anti-EGFR or anti-IGF1R resistance (see Table 1). Although the activity of the receptor (IGF1R phosphorylation) would be a solid biomarker, the similarity of IGF1R with INSR complicates the application of this approach in clinical practice. A pIGF1R antibody (anti-pY1316, a COOH-terminal antibody provided by Dr. Rubini) was used by our group to address IGF1R activation. Curiously, nuclear pIGF1R staining (which occurs in less than 10% of patients), but no other more common patterns of staining (such as cytoplasmic or membrane-associated), is correlated with chemotherapy and targeted therapy resistance [68].

Our group and others have evaluated potential IGF biomarkers based on liquid biopsy (ligands (IGF or IGFBP) by ELISA) [69,70,71,72] or IGF, IGFBP, and/or IGF1R protein in paraffin-embedded tissues (immunohistochemistry [73,74]), DNA (polymorphisms [10]), or RNA expression [75]. Global interpretation is hampered by trial design (most results are derived from retrospective analysis that were not designed for this purpose), nonstandardized biomarker cut-offs, and the use of different techniques (ELISA, DNA polymorphisms, and RNA expression). However, we conclude that high levels of IGFBP could identify a subgroup of mCRC patients with better prognosis [72]. It remains unclear whether any of these markers would predict the efficacy of IGF1R or EGFR inhibitors.

### 3.2. Liquid Biopsy in RAS Wild-Type Anti-EGFR Pretreated Patients to Select Optimal Patients for New Therapies. Mechanism of Resistance

The identification of biomarkers of the response to anti-EGFR therapy has proven to be promising. The monoclonal antibodies cetuximab and panitumumab, which neutralize the extracellular domain of EGFR, have demonstrated effectiveness, in terms of both a better response and improved survival when used as first-, second-, or third-line treatment regimens. Unfortunately, however, patients with mCRC who have mutant RAS (KRAS/NRAS; ~55%) or BRAF (V600E; ~5–10%) genes do not derive any therapeutic benefit from anti-EGFR drugs [76,77]. Although patients with wild-type (WT) RAS and BRAF (30–35% of mCRC patients) are sensitive to first-line chemotherapy plus anti-EGFR treatment (best overall response (BOR) range, 55–75%) [78,79,80], most of these patients develop resistance within the first 2 years. Primary inter- and intratumor heterogeneity and acquisition of secondary RAS and BRAF mutations during the course of such treatments are the major factors responsible for acquired resistance [81,82,83,84,85,86,87,88] (Figure 1). The recent gene expression-based molecular classification of consensus molecular subtypes (CMSs) has demonstrated the existence of phenotypic heterogeneity in CRC [89], highlighting the need for improved risk stratification and selection of patient populations before any specific treatment options are offered. Although the CMS classification in mCRC suggested prognostic significance, its clinical significance in terms of predicting the response to anti-EGFR therapy remains unclear [90,91,92]. Furthermore, given that genetic mutations can induce certain gene expression phenotypes in CRC, a comprehensive phenotypic exploration (including not only cancer epithelial cells, but also the tumor cell-type microenvironment) is critically required to establish biomarkers that can robustly predict the response to anti-EGFR therapies.

### 3.3. New Targeted Agents in Anti-EGFR Pretreated Patients with RAS Wild-Type mCRC

#### 3.3.1. Selection of Optimal Patients with Liquid Biopsy (RAS and BRAF)

CRC patients included in clinical trials of anti-IGF1R compounds and Sym004 (a mixture of two monoclonal antibodies—futuximab and modotuximab—that binds epitopes in the EGFR extracellular domain) were not selected based on RAS/BRAF WT in liquid biopsy [8,71,93,94]. This aspect is important because 30–35% of included patients basally had RAS or BRAF mutations that are related to intrinsic EGFR resistance [94]. Therefore, we propose that new agents or re-challenge strategies in patients progressing to anti-EGFR include only patients with liquid biopsy determination of the RAS and BRAF genotype. In ongoing EGFR re-challenge studies or those with new compounds (anti-HER2 and anti-MET), liquid biopsy is not mandatory to exclude RAS/BRAF patients. We believe that this limitation can compromise the efficacy of new drugs.

#### 3.3.2. Prospective Trials with New Strategies or New Compounds (HER2 Inhibitors, MET Inhibitors, and Re-Challenge with EGFR Inhibitors [Cetuximab or Panitumumab])

The IGF axis has been proposed as a target for anticancer therapies. Antibody, tyrosine kinase, and ligand inhibitors of the IGF receptor have been studied. In phase I trials, these antibodies seem to be well-tolerated; the most common toxicity is hyperglycemia [95]. Early-phase trials in the last lines of treatment showed promising results, mainly a stable response, but the results were negative in later-phase trials. The efficacy of anti-IGF1R and Sym004 was hugely disappointing, with nearly null activity with anti-IGF1R compounds (ganitumab, figitumumab, and dalotuzumab) and less than a 15% response rate with Sym004 [71,74,93,94]. One of the major limitations in these studies is the absence of targeted biomarker selection. This is a potentially important issue to address with new strategies (re-challenge with EGFR compounds [96]) or new agents (HER2 and MET inhibitors). Re-challenge strategies lack a suitable biomarker for patient selection, but only patients with HER2+++ overexpression (with DS8201) or patients with MET amplification should probably be included with new agents. The importance of biomarker selection has been highlighted by the low activity of MET inhibitors without biomarker selection [97,98].

A second important area concerns the trial design itself. Although objective responses have been reported with re-challenge anti-EGFR strategies involving cetuximab and irinotecan, the overall response rate is less than 15% [96]. It is important to note that, in this trial, it is unclear if the efficacy was due to irinotecan, cetuximab, or their combination. Regardless, the benefit in the subgroup of patients without RAS/BRAF mutation seems very modest (median progression-free survival (PFS), 4.1 months). We speculate that the efficacy of new strategies (e.g., re-challenge with anti-EGFR) or new compounds would probably not deserve further phase III trial development if monotherapy activity is below 30%, which is the expected response rate in the CHRONOS trial. In randomized clinical trials with standard-of-care schedules in second-line therapies (e.g., FOLFIRI in the BEYOND trial) or third-line therapies (regorafenib or TAS-102 in the PULSE trial, TAS-102 in the VELO trial, or regorafenib or investigator choice in the FIRE-4 trial), significant differences in PFS or overall survival (OS) should be calculated based on ESMO-MCBS expectations.

Five clinical trials have reported HER2 inhibitor activity in pretreated double wild-type *RAS* and *BRAF* mutation (2WT) mCRC patients with HER2 overexpression or amplification [99,100,101,102,103] (Table 2). Four phase II trials have evaluated combinations of trastuzumab with lapatinib [99], trastuzumab plus pertuzumab (100,101), or trastuzumab/T-DM1 plus pertuzumab [102]. Response rates ranged from 9% to 35% with a modest median PFS of 4 months. The only reported phase II study with trastuzumab-deruxtecan (DS-8201a), DESTINY-CRC 001, showed a promising 45% BOR and median survival of 7 months [103]. DS-8201a is a HER2-targeting antibody–drug conjugate, structurally composed of a human anti-human HER2 antibody, an enzymatically cleavable peptide linker, and a topoisomerase I inhibitor (DX-8951). The results of the DESTINY-CRC 001 are in accordance with preclinical data showing that DS-8201a has improved antitumor efficacy compared with T-DM1 [104]. In view of these results, the development of DS-8201a in HER2-positive mCRC patients has been prioritized. The DESTINY-CRC03 clinical trial will compare DS-8201a vs. standard of care in second-line 2WT CRC patients with HER2+++ overexpression.

MET inhibitors (capmatinib and tivantinib) have been tested in combination with cetuximab in two phase II studies [97,98]. Globally, in nonselected patients, the activity is very modest with a BOR < 10% and a less than 3-month median PFS. An unplanned subanalysis with tivantinib suggested that MET-amplified patients would be more sensitive to MET inhibition. Due to this subanalysis, a prospective clinical trial has recently begun recruiting mCRC patients with MET amplification (PERSPECTIVE). Although this trial will include only mCRC patients with MET amplification, other well-known resistance mechanisms such as EMT (which increases not only hepatocyte growth factor, but also other alternative EMT-driven biomarkers) would also contribute to tepotinib-cetuximab resistance [105].

#### 3.3.3. EGFR and Trastuzumab-Acquired Resistance

In addition to genetic mutations, a comprehensive profiling of gene expression patterns and their application in tissue and plasma samples might not only allow identification of superior predictive biomarkers for anti-EGFR therapy, but also further understanding of potential new strategies (such as re-challenge with anti-EGFR) or the addition of new agents. Indeed, less than 30% of patients that progress to chemotherapy plus anti-EGFR compounds can currently be explained by genetic resistance mechanisms and detected by ctDNA analysis. A transcriptomic resistance mechanism to cetuximab has recently been described, emphasizing the importance of the immune microenvironment in anti-EGFR resistance [88]. Cetuximab and panitumumab exert at least part of their activity through Fc-mediated antibody-dependent cellular cytotoxicity (ADCC), although this mechanism of resistance has largely been unexplored. ADCC activity is hampered by M2 microenvironment polarization [106,107] or antibody-dependent cellular phagocytosis (ADCP) loss mediated by dendritic cells [108] and, therefore, deciphering immune microenvironment changes could potentially improve current anti-EGFR efficacy.

Trastuzumab activity in preclinical models has recently been associated with ADCP mediated by macrophages but not with ADCC through natural killer cells or neutrophils, complement cellular cytotoxicity (CDC), or T-cell adaptive immunity [109]. CD47 overexpression in cancer cells (a “do not eat me” receptor) has been inversely associated with trastuzumab efficacy. DS-8201a, also in a preclinical CRC mouse model, upregulates CD86, dendritic cells, and CD8 T-cells and PD-L1 and MHC-I in tumor cells, which emphasizes the importance of not only ADCP, but also adaptive immunity in drug activity. It seems that its crucial effect on immune activation is specifically due to the effect of deruxtecan [110]. Despite these preclinical data with trastuzumab and DS8201a, only 45% of patients clinically responded to DS8201a and less than 30% to trastuzumab plus lapatinib or pertuzumab. In addition, most initial responders (>90% of cases) progress in the first year. Therefore, understanding of how metabolic reprogramming can impair ADCP function or how chronic therapeutic antibody-mediated ADCP stimulation can influence acquired mechanisms of trastuzumab or DS8201a resistance would be valuable. Autophagy, a metabolic reprogramming process increased in highly hypoxic nutrient-deprived tumors such as pancreatic adenocarcinomas [111], has recently been shown to increase immune suppression [112,113]. It is further unclear how autophagy suppresses the immune system, although Yamamoto et al. suggest that autophagy increases immunosuppression through reduced MCH-I expression [112]. Other authors indicate that autophagy can decrease TNFα-dependent cell death by immune CD8+ T-cells [113]. We are tempted to speculate that continuous ADCP stimulation (either by cancer-mediated autophagy metabolic reprogramming or antibody-mediated stimuli) induces LAP-dependent M2 polarization [114] and PD-L1 and IDO-1 expression [115] (see Figure 2).

## 4. Conclusions

The structure and signaling of IGF1R suggest that targeted therapies should be effective if they disrupt this oncogenic pathway. However, alternative receptor tyrosine kinases (EGFR, MET, HER2), post-translational processes (such as SUMOylation), IGF1R-EMT-independent pathways, and cancer–cell microenvironment interactions suggest that current preclinical models do not fully recapitulate the complex clinical scenario. This can explain why, even though IGF1R targeting in preclinical models showed high activity in CRC and across tumor types, IGF1R inhibition failed in clinical trials. Additional reasons for the lack of anti-IGF1R efficacy in mCRC include nonselection of patients based on RAS/BRAF WT liquid biopsy.

A comprehensive analysis that should include not only characterization of intrinsic malignant epithelial mechanisms of resistance (genomic (e.g., RAS/BRAF), non-genomic (e.g., SUMOylation), and EMT), but also nonepithelial malignant microenvironment-dependencies should be applied to evaluate new strategies with old drugs (anti-EGFR re-challenge with cetuximab or panitumumab) or new agents (HER2 and MET inhibitors) (see Figure 3). In addition, all of these aspects should also be re-evaluated with novel combinations of targeted therapies involving IGF1/2 antibodies (cetuximab, trastuzumab, panitumumab, DS8201a) and checkpoint inhibitors or other immune therapies. The lessons learned with receptor tyrosine kinase inhibitors should improve our knowledge for adequate drug development in mCRC patients.

## Figures and Tables

**Figure 1 ijms-22-05019-f001:**
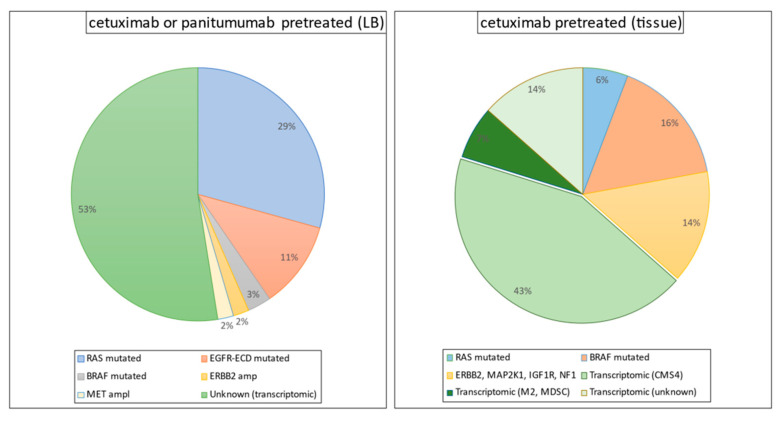
Mutation profile in biopsy and liquid biopsy (LB) samples from pretreated colorectal patients with KRAS, NRAS, and BRAF wild-type.

**Figure 2 ijms-22-05019-f002:**
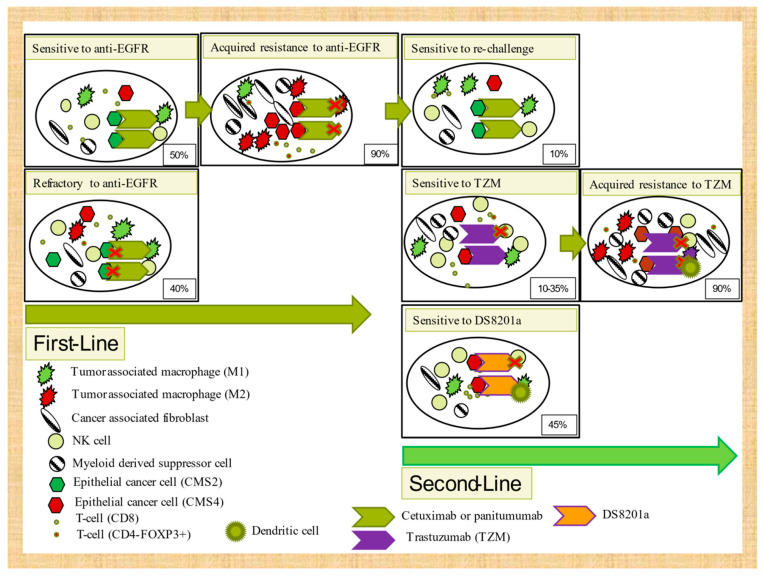
Tumor microenvironment mechanisms of intrinsic and acquired resistance to targeted therapies in 2WT RAS/BRAF colorectal cancer.

**Figure 3 ijms-22-05019-f003:**
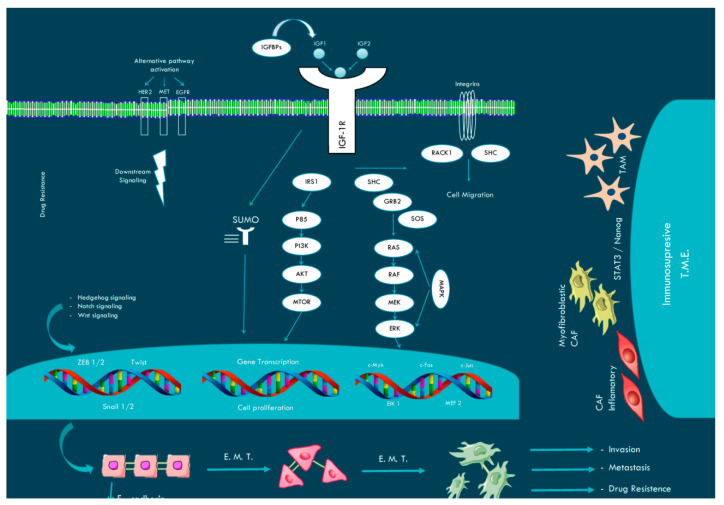
Schematic diagram of the IGF1R pathway. IGF1R phosphorylates the scaffold proteins IRS1 and can bind both PI3K (via the p85 subunit of PI3K) and the GRB2-SOS complex. PI3K activates downstream signaling (AKT, MTOR), whereas GRB-SOS stimulates RAS to activate the MAPK pathway (RAF/MEK/ERK). Both pathways can stimulate transcription factors such as c-Myc, c-Fos, and c-Jun to promote cell survival, proliferation, invasion, and metastasis. IGF1R crosstalks with integrins and RACK1 and FAK proteins and promotes cell migration. In addition, IGF1R SUMOylation and EMT can also contribute to proliferation and apoptotic resistance. Interaction with other RTKs such as HER2, MET, and EGFR constitutes an alternative IGF1R pathway. Finally, IGF1R signaling can also promote an immunosuppressive tumor microenvironment.

**Table 1 ijms-22-05019-t001:** IGF1R pathway biomarkers in metastatic colorectal cancer.

Author	Study Design	Treatment Arms	LB/N	Biomarker Methodology	Biomarker	Conclusion
**Winder et al. [10]**	Retrospective/prospective	Cetuximab	No/130	Polymorphisms	IGF1 (rs2946834-AA)	Increased efficacy
**Codony-Servat et al. [47]**	Retrospective/prospective	CHT+/−mAb	No/470	IHC	High pIGF1R (nuclear)	Decreased efficacy
**Fuchs et al. [69]**	Retrospective/prospective	CHT	Yes/527	ELISA	High IGFBP3 and IGF1	Improved OS
**García-Albéniz et al. [70]**	Retrospective	CHT	Yes/41	ELISA	IGF1 increment	Improved OS
**Van Cutsem et al. [71]**	Retrospective/prospective	Ganitumab-Panitumumab vs Panitumumab	Yes/94	ELISA	High IGFBP1 and IGFBP2	Improved OS
**Guercio et al. [72]**	Prospective	CHT+mAb	Yes/1084	ELISA	High IGFBP3 and IGFBP7	Improved OS
**Scartozzi et al. [73]**	Retrospective	I-Cetuximab	No/112	IHC	Low IGF1	Increased efficacy
**Sclafani et al. [74]**	Retrospective/prospective	Dalotozumab-Cetuximab-I vs Cetuximab-I	No/344	IHC	High IGF1	Decreased OS
**Huang et al. [75]**	Retrospective	Cetuximab	No/70	RNA expression	High IGF1R	Increased efficacy

CHT, chemotherapy; LB, liquid biopsy; N, number of patients included; I, irinotecan; IHC, immunohistochemistry; mAb, monoclonal antibody; pIGF1R, phosphorylated IGF1R.

**Table 2 ijms-22-05019-t002:** New strategies in 2WT mCRC patients after anti-EGFR progression.

Author	Design(n)	Treatment arms	LLB at Entry *	Biomarker Driven **/Targeted Therapy BiBiomarker Driven **/Targeted Therapyomarker Selection	BOR%	PFS (m)	mOS (m)	ESMO-MCBS: ESMO-Magnitude Clinical Benefit Scale
**Montagut et al. [94]**	II-R (254)	Sym004-12 mg/kg vs. Sym004-9 mg/kg vs. Investigator choice	No	No	14.1 9.6 2.9	- - -	11.9 8.9 8.4	1
**Cremolini et al. [96]**	II (28)	Irinotecan-Cetuximab	No	No	14	4.1 **	-	NA
**Rimassa et al. [97]**	II (41)	Tivantinib-Cetuximab	No	No	9.8	2.6	-	NA
**Delord et al. [98]**	I-II (13)	Capmatinib-Cetuximab	No	No	0	-	-	NA
**Sartore-Bianchi et al. [99]**	II (27)	Trastuzumab-Lapatinib	No	Yes/HER2 (+++)	26	5.1	-	NA
**Nakamura et al. [100]**	II (18)	Trastuzumab-Pertuzumab	Yes	Yes/HER2 (amplification)	35	4	-	NA
**Gupta et al. [101]**	II (28)	Trastuzumab-Pertuzumab	No	Yes/HER2 (+++)	25	4.2	-	NA
**Sartore-Bianchi et al. [102]**	II (31)	TDM1-Pertuzumab	No	Yes/HER2 (+++)	9.7	4.1	-	NA
**Siena et al. [103]**	II (53)	Trastuzumab-Deruxtecan	No	Yes/HER2 (+++)	43.4	6.9	-	NA
**CHRONOS ***	II (129)	Panitumumab	Yes	No	-	-	-	-
**BEYOND ***	II-R (85)	FOLFIRI-Panitumumab vs FOLFIRI	Yes	No	-	-	-	-
**VELO ***	II-R (112)	TAS-102-Panitumumab vs TAS-102	No	No	-	-	-	-
**PULSE ***	II-R (106)	Panitumumab vs Regorafenib or TAS-102	Yes	No	-	-	-	-
**FIRE-4 ***	III (230)	Irinotecan-Cetuximab vs Regorafenib or Investigator choice	No	No	-	-	-	-
**PERSPECTIVE ***	II (48)	Tepotinib-Cetuximab	Yes	Yes/MET (amplification)	-	-	-	-

2WT, double wild-type for RAS and BRAF mutations; N, number; LB, liquid biopsy; BOR, best overall response; mPFS, median progression-free survival in months; mOS, median overall survival in months; ESMO-MCBS, ESMO-Magnitude Clinical Benefit Scale; NA, not applicable. * Prospective studies without available results. ** Patients without mutations in liquid biopsy. (+++) Amplification 3 crosses.

## Data Availability

Not applicable.

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
