# Peer review of "Lessons to Learn for Adequate Targeted Therapy Development in Metastatic Colorectal Cancer Patients"

_ijms, 2021, doi:10.3390/ijms22095019_

Round 1

Reviewer 1 Report

Minor comments

  1. In tables, the authors must write full terms but not abbreviation alone; e.g. I, MAB, C, P, 2WT, b, and so on.
  2. In Figure 1, gray ink is hard to read, thus, the authors should use clearer colors.
  3. In Figure 2, some words overlapped, thus they must write not be overlapped.
  4. In Figure 3, some words in the bottom were missing, please write clear.

Author Response

Comments and suggestion from the first reviewer

Reviewer 1#

Minor comments

  1. In tables, the authors must write full terms but not abbreviation alone; e.g. I, C, P, 2WT, b, and so on.

I apologize to the reviewer. We have written full terms description of the abbreviation in the tables

  1. In Figure 1, gray ink is hard to read, thus, the authors should use clearer colors

Following reviewer suggestion, we have used clearer colors in Figure 1.

  1. In figure 2, some words overlapped, thus they must write not to be overlapped.

Following reviewer suggestion, we have written words without overlapping in Figure 2.

  1. In figure 3, some words in the bottom were missing, please write clear

Following reviewer suggestion, we have written in a clear way the Figure 3.

Reviewer 2 Report

ijms-1189022

Oliveres et al., IGF-1R pathway. Lessons to learn, for adequate future drug development in RAS WT metastatic colorectal cancer patients

 The paper of Oliveres et al. is presented as a review and aims to discuss new and alternative strategies in drug development for RAS WT mCRC patients with an emphasis on IGF-1R pathway and anti-IGF-1R therapy.

Unfortunately, the overall quality of this manuscript is not sufficient for a publication. There are many reasons for that. This article claims to be about IGF-1R pathway strategies, but the majority of the text deals with other hormones/receptors like EGFR. Moreover, the article completely ignores involvement of insulin/IR and its interference with IGF-1 signaling (see e.g. a recent review in doi.org/10.1038/s41568-020-0295-5). The graphical quality of Figures is very low and the text in Figures is not readable. Both tables look as to be copied from another paper, including citations. Tables completely lack explanation of symbols, data, values etc. The English of the paper is not very good and the text needs editing by a native speaker. Citations in (at least) the first chapter (IGF-1R structure and function) are not well chosen as they are either old or inappropriate. It is not true that IGF-1R mutations have not been described (look at doi: 10.1530/EC-17-0038 or DOI: 10.1056/NEJMoa010107). And I could continue for a long time in my criticisms.

I think that the manuscript should be rejected and reconsidered only after complete revisions.

Author Response

Comments and suggestion from the second reviewer

Reviewer 2#

  1. This article claims to be about IGF-1R pathway strategies, but the majority of the text deals with other hormones/receptors like EGFR.

We agree with the reviewer that the third part of our review includes other pathways and not only IGF-1R compounds, that are currently on development in double wild-type metastatic colorectal cancer but the first two parts are dedicated to IGF axis. The reason is because IGF-1R inhibitors failed in this area and currently EGFR compounds are the only available for use in clinical practice. Anyway, and considering the reviewer comment we have expanded other important aspects in the IGF axis that we miss in the first version of our manuscript.

  1. Moreover, the article completely ignores involvement of insulin/IR and its interference with IGF-1 signaling (see e.g. a recent review in doi.org/10.1038/s41568-020-0295-5).

We thanks to the reviewer for this appointment. We have incorporated in the point 1 (IGF-1 axis) these references and we have emphasized the importance of the insulin/IR axis in the IGF1R resistance also in the point 2.1 of our revised version.

  1. The graphical quality of figures is very low and the text in figures is not readable.

We have improved the quality of the figures and the text.

  1. Both tables look as to be copied from another paper including citations. Tables completely lack explanations of symbols, data, values, etc.

Both tables are original tables. We apologize the reviewer regarding explanations of symbols, data, values, etc. We have incorporated explanations of symbols, data, values, etc in the new version of the tables.

  1. Citations in (at least) the first chapter (IGF-1R structure and function) are not well chosen as they are either old or inappropriate.

In accordance with reviewer suggestion, we have changed practically all the references in the first chapter.

  1. It is not true that IGF-1R mutations have not been described (look at doi:10.1530/EC-17-0038 or DOI: 10.1056/NEJMoa010107).

I apologize again at that point. We would emphasize that IGF-1R mutations are not described in human tumors. We have clarified this point in the revised version and we have included the two references also.

Reviewer 3 Report

This manuscript reviews the role of IGF-IR in RAS WT metastatic colorectal cancer and the lessons that could be learnt to improve drugs to target the pathway.

The title and the described aim of the manuscript do not accurately reflect the content of the manuscript. The manuscript starts with a reasonably standard concise introduction to the IGF-IR pathway. A large amount of the text, and that which is more novel then deals with the results of clinical trials of other targeted therapies, targeting EGFR, HER2 and MET. It would be more appropriate to include in the title that the manuscript describes the lessons learned from other targeted therapies that could inform more effective targeting of the IGF-IR pathway.

Other specific points.

  1. In the abstract it is stated that: “different AKT/MEK independent pathways (such as IGF-1R SUMOylation) and epithelial-mesenchymal transition (EMT) can also contribute to proliferation and apoptotic resistance”. SUMOylation is a post-translational modification of proteins and EMT is a process of cell differentiation: these are not pathways and this should be reworded to avoid confusion.
  2. On page 2, line 67: it is stated that “IGFBPs are tightly regulated by metalloproteinases”; it should be mentioned that other proteinases, particularly serine proteases are also involved in this process.
  3. On page 2, section 2.2 focuses on IGF-IR SUMOylation, which is described in detail. There is also mention of acetylation, ubiquitination and phosphorylation. It is not clear why there is such emphasis on SUMOylation when these are also post-translational modifications that can modulate the activity of the pathway; there is not evidence that SUMOylation is the most prevalent or the most important modifier. Indeed, when considering the pathway as a whole, the balance of tyrosine-phosphorylations to serine- and threonine-phosphorylations has been most studied and this is not mentioned. There should be a better balance of considerations of modulators of the pathways by post-translational modifications.
  4. There are many acronyms used throughout and several are not defined; they should be defined at first use. Such as: SYM004 and ADCC.
  5. On page 2, line 86: SUMOylation is described as: “can covalently and reversibly attach to the lysine of a substrate protein, causing post-translational modification of a protein”. This is misleading as SUMOylation is a form of post-translational modification and not a cause of post-translational modification.
  6. Page 3, Line 114: the sentence indicating that “when IGF-IR is phosphorylated in the nucleus, currently is undruggable” needs rewording and citations need to be added, both for IGF-IR being phosphorylated in the nucleus (as opposed to phosphorylated IGF-IR being translocated into the nucleus) and for it being undruggable. Most small molecule tyrosine kinase inhibitors can get into the cell (the kinase domain of the cell surface receptor is on the intracellular side) and potentially get into the nucleus: evidence that they do not stop phosphorylation of IGF-IR in the nucleus needs to be cited.

Minor points:-

There are numerous typographical and grammatical errors that need correcting. These include:-

Page 1, line 36: ‘has’ should be ‘have’.

Page 2, line 83: the ‘a’ should be removed from “mainly observed as a nuclear and peri-nuclear proteins”.

Page 3, line 107: ‘it’ is needed in “because it acts as a…”

Page 3, line 108: should be: “and its nuclear location…”

Page 3, line 109: should be ‘have’ and not ‘has’.

Page 3, line 113: should be ‘decreased’ not ‘decrease’.

Page 3, line 128: “combine between other” needs rewording.

Page 4, line 150: “greatly difficult it’s use” needs rewording.

Page 4, line 152: ‘tinction’ is not a commonly used term and to avoid confusion a more generally used term such as ‘staining’ should be considered.

Page 9, line 304: ‘if disrupt’ should be ‘if they disrupt’.

Page 9, line 318: should be “IGF1/2 antibodies”.

Author Response

Comments and suggestion from the third reviewer

Reviewer 3#

The title and the described aim of the manuscript do not accurately reflect the content of the manuscript. The manuscript starts with a reasonable standard concise introduction to the IGF-1R pathway. A large amount of the text, and that which is more novel the deals with the results of clinical trials or other targeted therapies, targeting EGFR, HER2 and MET. I would be more appropriate to include in the title that the manuscript describes the lessons learned from other targeted therapies that could inform more effective targeting IGF-1R pathway

We agree with the reviewer comment. In fact, lessons to learn are bidirectional and concerns to anti IGF-1R compounds but also to other targeted agents. Therefore, we have changed the title to include a more general consideration.  Now the title is ‘’Lessons to learn, for adequate targeted agent development in metastatic colorectal cancer patients’’

Other specific points

  1. In the abstract it is stated that: ‘’different AKT/MEK independent pathways (such as IGF-1R SUMOylation) and epithelial-mesenchymal transition (EMT) can also contribute to proliferation and apoptotic resistance’’. SUMOylation is a post-translational modification of proteins and EMT is a process of cell differentiation: there are not pathways and this should be reworded to avoid confusion.

We fully agree with the reviewer appointment and apologize for the confusion. We have completely revised the abstract in the new version, following reviewer recommendations.

  1. On page 2, line 67: it is stated that ‘’IGFBPs are tightly regulated by metalloproteinases’’; it should be mentioned that other proteinases, particularly serine proteases are also involved in this process.

Following reviewer recommendation this phrase has been revised to include also serine proteases in the IGFBP regulation.

  1. On page 2, section 2.2 focuses on IGF-1R SUMOylation, which is described in detail. There is also mention of acetylation, ubiquitination and phosphorylation. It is no clear why there is such emphasis on SUMOylation when there are also pots-translational modifications that can modulate the activity of the pathway. Indeed, when considering the pathway as a whole, the balance of tyrosine-phosphorylations to serine-and threonine-phosphorylations has been most studied and this is not mentioned. There should be a better balance of considerations of modulators of the pathways by post-translational modifications.

We recognized a bias with SUMOylation instead of other post-translational processes such as acetylation, ubiquitination and phosphorylation, due to in part our previously published work in the area was focus on SUMOylation (Codony-Servat, BJC, 2017). Anyway, and following reviewer suggestion, we have changed the title of this section that includes other post-translational processes. Other post-translational processes like phosphorylation are partially revised in sections 1 and 2.1.

  1. There are many acronyms used throughout and several are not defined; they should be defined at first use. Such as; SYM004 and ADCC.

We have revised all the acronyms and defined at first use

  1. On page 2, line 86: SUMOylation is described as ``can covalently and reversible attach to the lysine of a substrate protein, causing post-translational modification pf a protein’’. This is a misleading as SUMOylation is a form of post-translational modification and not a cause of post-translational modification.

I apologize for the misleading phrase. We have reworded adequately this sentence.

  1. Page 3, Line 114: the sentence indicating that ‘’when IGF-1R is phosphorylated in the nucleus, currently is undruggable’’ needs rewording and citations need to be added, both for IGF-1R being phosphorylated in the nucleus (as opposed to phosphorylated IGF-1R being translocated into the nucleus) and for it being undruggable. Most small molecule tyrosine kinase inhibitors can get in the cell (the kinase domain of the cell surface receptor is on the intracellular side) and potentially get into the nucleus: evidence that they do not stop phosphorylation of IGF-1R in the nucleus needs to be cited.

This is a very interesting observation. Potentially IGF1R tyrosine kinase inhibitors can inhibit membrane phosphorylation and therefore nuclear phosphorylation, but we have observed that both monoclonal antibodies (ganitumab) and IGF1R tyrosine kinase inhibitors (AEW541) contradictorily, increase IGF-1R in the nucleus thorough PIAS3 increment. We have reworded this sentence in the revised version.

Minor points

There are numerous typographical and grammatical errors that need correcting. These include…

All these typographical and grammatical errors have been revised by a native speaker